# Design and Test of the Clearing and Covering of a Minimum-Tillage Planter for Corn Stubble

**Shouyin Hou, Shengzhe Wang, Zhangchi Ji and Xiaoxin Zhu \***

College of Engineering, Northeast Agricultural University, Harbin 150030, China
* Correspondence: zhuxiaoxin@neau.edu.cn

**Abstract:** Conservation tillage technology can reduce wind erosion and soil erosion, improve soil fertility, avoid straw burning and relieve ecological pressure. It is an important measure to achieve sustainable agricultural development. In northeast China, there is a large amount of straw covering the ground after the corn machine harvest, which can easily lead to the blockage of the soil-touching parts during no-tillage seeding, affecting sowing quality and crop yield. In order to solve the above problems, the clearing and covering of a minimum-tillage planter for corn stubble was developed. The machine can complete multiple processes, such as seedbed preparation, seeding, fertilization, covering and suppression, straw covering, etc., in a single entity. This paper focuses on the design of the straw cleaning device and uses discrete element method software (EDEM 2018, Altair Engineering, Troy, MI, USA) to establish the straw cleaning device–straw–soil discrete element simulation model. The quadratic-regression orthogonal center-of-rotation combination test method is used to optimize the parameter combination of the machine, using the operating speed, the speed of the knife roller and the penetration depth of the knife as the test factors and using the rate of cleaning straw and the equivalent power consumption as the evaluation index. The results show that each factor has a significant influence on the performance evaluation indices, and the order of influence of each factor on the rate of cleaning straw is operation speed > penetration depth of knife > speed of knife roller, and the order of influence of each factor on the equivalent power consumption is penetration depth of knife > speed of knife roller > operation speed. The optimal combination of parameters is a 5.5–6.2 km/h operation speed, a 500 rpm speed of the knife roller, a 40 mm penetration depth of the knife, a straw-cleaning rate of more than 90% and an equivalent power consumption of less than 8 kW. This study provides technical and equipment support for the promotion of conservation tillage technology in Northeast China.

**Keywords:** conservation tillage; minimum-tillage seeding; straw clearing and covering; discrete element method; design; test



## 1. Introduction

Conservation tillage technology is a new agricultural tillage system and technology system with soil health as the core, which is based on the methods of minimum-tillage, no-tillage, surface micro-topography transformation and direct sowing on straw mulched farmland. It aims to increase soil organic matter content and construct a fertile tillage layer and has the advantages of water storage, improving soil structure, reducing soil erosion, improving soil fertility and increasing crop yield [1,2]. In northeast China, where corn is grown on a large scale and the degree of agricultural mechanization is high, combine harvesters are usually used to harvest and return the full amount of corn stover to the ground by shredding. Due to the large production of corn stover in the region [3,4], no-tillage sowing is prone to the clogging of the touching parts, difficulty in getting the furrow opener into the soil, straw pressing into the seed furrow and sowing monolithic bias, resulting in low operating speed, long maintenance time, uneven sowing grain spacing and depth, the seed being hollowed out by straw, and poor covering and suppression. It seriously affects

the production efficiency and operation quality and limits the promotion and application of no-tillage sowing mechanization technology in northeast China [5]. In addition, China's existing no-tillage seeders have high power consumption, high operating costs, serious soil-compaction damage, poor anti-erosion effect, small machine operating width, and a low operating speed. This situation restricts the implementation of conservation-tillage-scale no- and minimum-tillage seeding in northeast China, results in energy waste and extended farming time and seriously endangers national food security production and sustainable agricultural development [6]. Therefore, the research and development of wide-width, high-speed, and no- and minimum-tillage seeders with high efficiency and anti-blocking function is the key to ensuring the realization of large-scale mechanized seeding and fertilization in areas with heavy straw cover [7].

No and minimum tillage is the core of conservation tillage technology, and anti-blocking is the key to no- and minimum-tillage seeding. Anti-blocking technology is a mechanized technology that adopts an anti-blocking device to prevent straw and weeds from clogging the seeding touching parts, which can be divided into two types: passive and active, according to the power source. Chen et al. [8] designed an offset double-disc trenching device, which provides downward pressure to two different-diameter trenching discs through hydraulic cylinders to achieve an anti-clogging effect and reduce the counterweight of the machine. Sharipov et al. [9] designed a no-tillage sowing-depth intelligent control planter, which controls the dynamic anti-blocking of the slide shoe furrow opener using the sensor recognition of the ground information. Lin et al. [10] designed a cutting and plucking anti-blocking device suitable for thick root stubble, and a serrated disc stubble-breaking plow blade based on the Archimedean spiral type made up for the slippage defect of the common disc blade, improved the root-stubble cutting rate and reduced power consumption. This kind of anti-blocking operation method causes little soil disturbance, has a simple device structure and usually relies on the self-weight of the machine or the pressure provided by the hydraulic system to complete the straw-clearing and stubble-breaking operation. However, the stubble-breaking and anti-blocking effects are poor when operating in plots with high soil firmness and many hard stubble crops, which affect the sowing quality.

Karayel [11] designed a combined wing shovel, a double-disc anti-blocking device that can effectively invade the untilled soil and place seeds at the appropriate depth. Wang et al. [12] designed a concave-disc passive monopod cleaning device, which uses the flip and fling action of the concave disc to dispose of weeds, broken stalks and root stubble cut by the flat-disc knife into the monopod furrow. Cao et al. [13] designed a side-cutter and stubble-cleaning-disc combination stubble-clearing device by notching the stubble-cutting disc to cut the straw on one side of the seed belt, using the stubble-cleaning disc to set the straw away from the seed belt and using the topsoil of the seed belt to bury and cover the straw, reducing the phenomenon of straw being blown back to the seed belt after being sown under the action of wind. This kind of anti-blocking-technology principle uses the friction with the ground to drive the anti-blocking device at low-speed to divert straw weed from the seed belt to a single side or to both sides to complete the clearing of straw; in certain conditions this process shows a good anti-blocking effect, when the amount of straw covered by the ground is large, making it prone to the blocking phenomenon in this scenario.

To avoid the blockage and pile dragging that tend to happen with passive anti-blocking devices when a large amount of straw covers the ground at high speed, the researchers also did a lot of research on active anti-blocking technology. Huang et al. [14] designed a combined stubble-cutting and straw-guiding straw–soil-separation device, which realized the process of cutting, breaking, tossing and guiding straw and weeds by combining driven stubble-cutting and passive straw-guiding, alleviating the problem of straw and weeds backfilling into the seed belt and mixing with straw and soil. Zhang et al. [15] designed a driving disc anti-blocking device, which improves machine passability and reduces the amount of soil disturbance by using the disc knife embedded in a combined trencher for

joint stubble cutting. The principle of such an anti-blocking device is to drive the disc knife through the tractor power-output shaft to cut the soil-surface straw and root stubble at high speed to break them up, but the power consumption of the knife is large.

Shi et al. [16] designed a full-volume straw-crushing strip-spreading and-seed belt-fractionation straw-clearing device, using the high-speed reverse-rotation of the straw-crushing knife to pick up the crushing function to form a straw-flow that fills the type cavity, creating straw-free seeding conditions for high-quality smooth no-tillage seeding. Ji et al. [17] designed a rototill-stubble bionic blade and compared it with the national-standard rototill knife and common stubble knife; the bionic blade's power consumption per unit of soil cutting area is less than the common stubble knife. Zhao et al. [18] designed a banded deep-pine stubble cutter, which can adapt to a land-preparation operation after autumn harvest or after straw treatment before planting, and improved the machine's smoothness and stubble-cutting efficiency. This anti-stubble technology means that the anti-stubble tool does not touch the surface soil and that it rotates at high speed in a shallow rotary state to crush the straw and root stubble to clear the relatively clean seeding area, and there are problems such as high energy consumption, the loud noise of the machine and the broken straw tending to backfill the seed bed.

In summary, for large-scale production in northeast China, there is an urgent need for a high-quality, high-efficiency, environmentally friendly, multifunctional mechanized seeding technology to implement mechanized conservation tillage no- and minimum-tillage seeding under the premise of ensuring crop yield, improving operational efficiency and reducing operational energy consumption, to achieve the high-yield, high-efficiency and green sustainable development of agriculture in the region. This paper develops a clearing-and-covering minimum-tillage planter for corn stubble (hereinafter referred to as a straw-clearing and covering planter) with lateral straw-clearing and anti-blocking, precision sowing, lateral deep fertilization and straw-mulching integrated functions. Its structural components and operating principles have been introduced, and the lateral straw-cleaning and anti-blocking device (hereinafter referred to as the straw-cleaning device) has been designed. During the design process, the main parameters affecting the working performance of the device have been confirmed. A discrete element simulation model for the straw-clearing device–straw–soil under full corn-straw-mulching conditions has been established, based on EDEM software. A combination of virtual simulation and a quadratic-regression orthogonal center-of-rotation combination test has been used to determine the optimized combination of parameters affecting the working performance of the straw-removal device, and the simulation optimization results have verified through field tests.

## 2. Materials and Methods

### 2.1. Design of the Elastic-Tooth-Type Lateral Straw-Clearing Roller

2.1.1. Structure Composition and Working Principle

Figure 1a,b show the structural components of the straw-clearing and covering planter. The planter is composed of a straw-clearing device, fertilizer-application system, seeding unit and so on. The planter is hooked up to the tractor through a three-point suspension (8), and the tractor provides the traction force and the rotating power of the straw-cleaning-knife roller (10). Universal coupling is used to link the power-output shaft of the tractor and the power-input shaft (3) of the no-tillage planter and, through the knife-roller drive system, (2) to drive the eight transversely evenly arranged components in front of the sowing-unit straw-cleaning knife roller's (10) four components with symmetrical rotation on the left and right. The fertilizer discharger (4) and seed discharger (13) is driven by the ground wheel (5) through the chain drive.

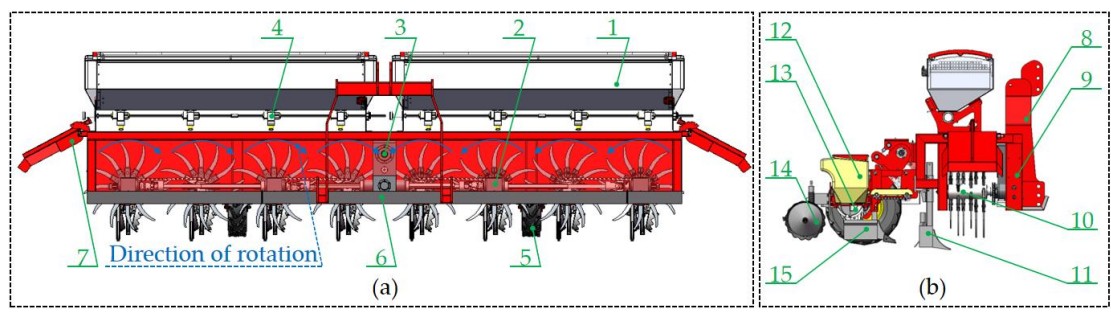

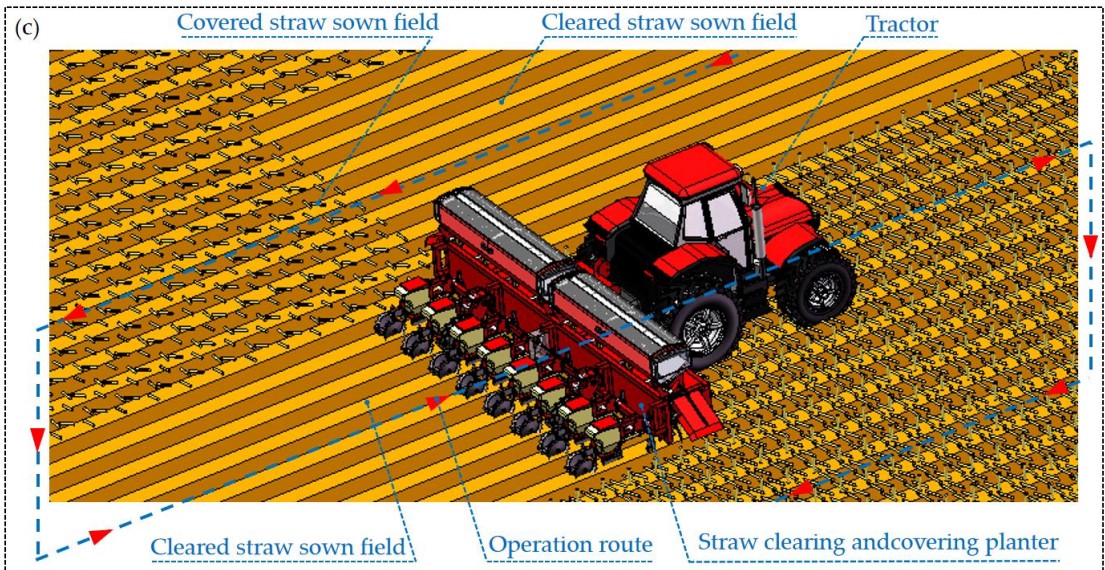

**Figure 1.** (**a**) Main view of the straw-clearing and covering planter: 1. fertilizer box, 2. knife roller transmission system, 3. power-input shaft, 4. fertilizer discharger, 5. ground wheel, 6. straw pressure plate, 7. straw barriers. (**b**) Side view of the straw-clearing and covering planter: 8. three-point suspension, 9. frame, 10. straw-cleaning knife roller, 11. fertilizer opener, 12. seedbox, 13. seed discharger, 14. mulching-suppression wheel, 15. seed opener. (**c**) Working-principle diagram of the straw-clearing and covering planter.

The working principle of the straw-clearing and covering planter and its technical characteristics are shown in Figure 1c. The tractor pulls the machine forward; the straw pressure plate introduces the straw into the working area of the straw-clearing device. The left and right groups of straw-cutter rollers constitute a 4-stage lateral-migration-and-throwing system of straw and stubble in the corresponding straw belt, which migrates and throws the straw and stubble in the corresponding straw belt step by step and clears the seedbed without straw cover and stubble residue. The fertilizing furrow opener and the sowing furrow opener open ditches and complete fertilization and sowing operations, covering the soil suppression of the seeding belt after the timely covering of soil and moderate repression and resulting in the formation of a clear straw-sowing ground. During the return operation, the straw-cutter roller will throw the straw from the operation to the surface of the first stroke of the clear straw sowing, complete the straw-covering operation, form the covered straw-sowing field and repeat the cycle until the end of the no-tillage-sowing operation.

### 2.1.2. Key Component Design

The straw-cleaning device is the core component of the straw-clearing and covering planter, and its working performance directly affects the quality and operational efficiency of the seeding. The overall structural parameters and key components need to be designed systematically.

According to the agronomic requirements, the straw-clearing and covering planter are designed for precise single-grain sowing, and the sowing grain spacing and row spacing are 160 mm and 650 mm, respectively. In northeast China, corn planting adopts a ridge planting mode. To sow the seeds on the ridge, the straw-cleaning knife roller should not only clean the covered straw in the sowing belt but also remove the root stubble in the sowing belt to avoid the blocking of the fertilizer opener and the seeding opener during the sowing process. To remove corn root stubble, the straw-cleaning knife installed on the straw-cleaning knife roller needs to have a reasonable depth of entry into the soil. During the pre-test study, it was found that the root-stubble-removal rate can reach more than 90% when the straw-cleaning knife enters the soil to a depth of 50 mm [19], and to reduce the amount of soil disturbance, reduce power consumption and improve efficiency, the design depth of soil penetration is 30–50 mm. To improve the efficiency of fertilizer utilization, a side-deep-fertilizer-application method was used, the front and back distance between the fertilizer opener and the seeding opener was 100 mm, the horizontal distance was 100 mm and the vertical distance was 90 mm. Considering the width of each opener, the theoretical sweeping size of the opener in the horizontal direction is 200 mm, so the width of the straw-clearing knife formed on the ground surface should be greater than 200 mm, and the design used in this paper is 300 mm. The overall arrangement of the left half of the straw-cleaning knife roller is shown in Figure 2.

The corn straw is cleaned by the straw-cleaning-knife rollers while completing the lateral conveyance step by step. To prevent straw residue from clogging the seeding and fertilizing touching parts and to improve the efficiency of straw transportation, corn straw should be transported in time, and the straw should be laterally scattered to the next-level straw-knife-roller straw-bandwidth range (Figure 2a), and the radius of the rotation of the knife roller should meet:

$$\begin{cases} R^2 = (R-d)^2 + \frac{w^2}{4} \\ \frac{l}{2} \le R \le l + \frac{D}{2} \end{cases} \tag{1}$$

where $R$ is the knife roller turning radius, mm; $d$ is the penetration depth of the straw-cleaning knife, mm; $w$ is the width of the straw-cleaning belt, mm; $D$ is the diameter of the knife shaft, mm; $l$ is the monopoly distance, mm.

Corn machinery after harvesting has a straw-crushing size gap and uses longer straw that is easy to wind in the knife shaft, resulting in a straw-cleaning-device blockage and lower efficiency. The knife-shaft diameter needs to be:

$$D \ge \frac{s_{\max}}{\pi} \tag{2}$$

where $s_{\max}$ is the maximum length of the straw, mm.

By substituting the penetration depth of the straw-cleaning knife and the width of the straw-cleaning belt into Equation (1), the knife-roller turning radius is 390 mm. To ensure the working performance of the straw-cleaning device, the straw-cleaning-knife roller is installed in front of the sowing monomer, and the knife-roller spacing is 650 mm. Through statistical analysis, the longest straw length after mechanical harvest was 310 mm; from Equation (2) can be obtained the requirement that the knife-shaft diameter be no less than 99 mm. The diameter of the rotating tool shaft is designed to be 121 mm, according to the standard finished seamless steel tube size [20].

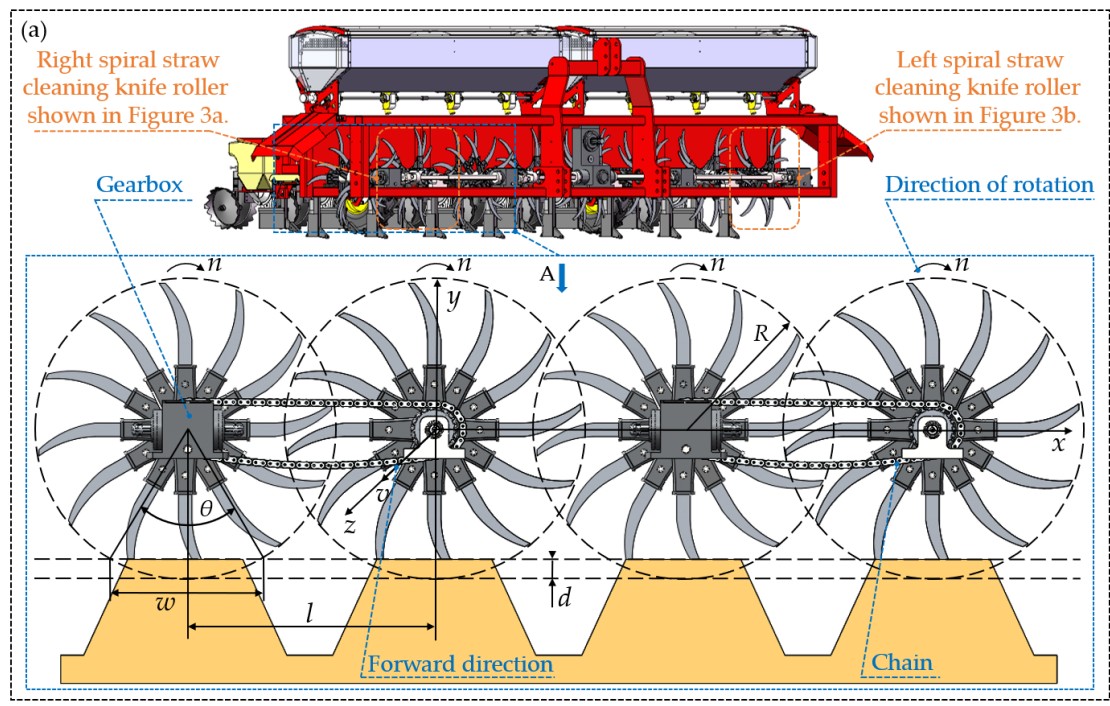

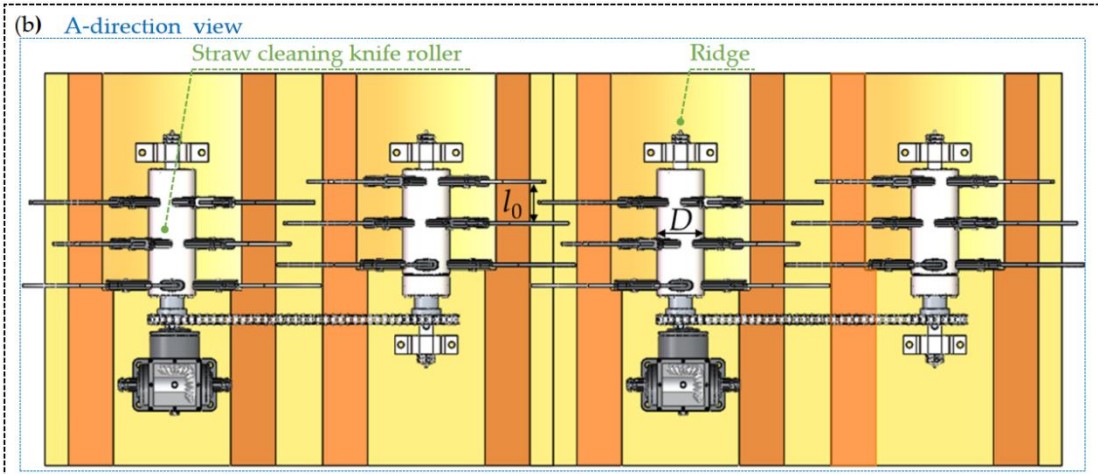

**Figure 2.** (**a**) The overall arrangement of the straw-cleaning knife roller. (**b**) Top view of the straw-cleaning knife roller.

The straw-cleaning-knife roller (Figure 3) is composed of a knife shaft, straw-cleaning knife, knife magazine, sprocket, and bearing. To improve the working quality and efficiency of the straw-clearing and covering planter and to reduce the power consumption and the vibration of the machine, the straw-cleaning knives need to be installed equally spaced in the circumferential and axial directions. The knife magazine is arranged around the knife shaft in the form of a spiral line, as observed by Figure 1a view; 4 groups of straw-cleaning-knife rollers on the left side are in a left spiral line, and 4 groups on the right side are in a right spiral line. At the same time, 8 groups of straw-cleaning-knife rollers are arranged symmetrically with the center line of the machine, and the straw is thrown on both sides.

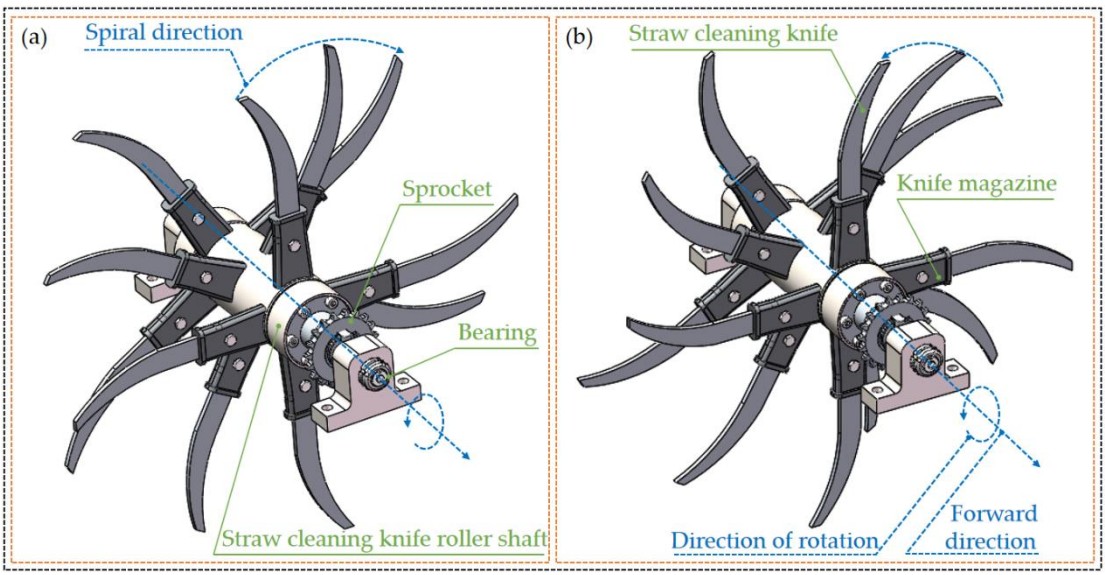

**Figure 3.** (**a**) The right spiral straw-cleaning knife roller. (**b**) The left spiral straw-cleaning knife roller.

As the straw-cleaning knife adopts a high-speed-impact cutting method to clean the straw, to reduce the vibration of the whole machine caused by its interaction with the soil, the design needs to ensure that each group of straw-cleaning-knife rollers has 1–2 straw-cleaning knives in contact with the soil at any time, and the relationship can be obtained:

$$\begin{cases} \frac{2\pi}{n_1 n_2} \leq \theta_{min} \\ \frac{4\pi}{n_1 n_2} \geq \theta_{max} \end{cases} \tag{3}$$

where $\theta_{min}$ is the minimum stubble-clearing angle, rad; $\theta_{max}$ is the maximum stubble-clearing angle, rad; $n_1$ is the number of straw-cleaning knives in the same rotary-plane circumferential direction; $n_2$ is the number of axial straw-clearing-knife rotary planes.

The known knife-roller radius of rotation and the penetration depth of the knife, through calculation, can be obtained to clear the stubble at an angle range of 0.79–1.02 rad; these factors can be used in Equation (3) to indicate straw-knife-roller need for the installation of 8–12 straw knives.

During the operation, the uniformity of the stubble-cutting pitch of the straw-cleaning-knife roller is the key to improving the stubble-removal rate; to meet the uniformity requirement, each parameter needs to satisfy Equation (4):

$$l_0 = 1 \times 10^5 \frac{v}{6nn_2} \tag{4}$$

where $l_0$ is the axial adjacent-straw-knife spacing, mm; $v$ is the operating speed, km/h; $n$ is the speed of the knife roller, rpm.

Through the team's preliminary experimental research [21], it was found that the operating speed of the machine was negatively correlated with the sowing quality of the sowing unit, and when the operating speed was greater than 7.2 km/h, the qualified rate of sowing depth and the qualified rate of sowing grain spacing would drop significantly. To meet the agronomic requirements and ensure maximum efficiency, the upper limit of the design operating speed is 7.2 km/h. Considering the large diameter of corn stubble and many roots in northeast China, the stubble-clearing effect is not ideal when the knife-roller speed is lower than 400 rpm, and it is difficult to realize the relative cleanliness of the area to be sown [22]. The knife-roller speed was initially selected to be 500 rpm. To avoid the blockage of the adjacent straw-cleaning knife with corn stubble, the axial spacing between adjacent straw-cleaning knives on the straw-cleaning roller is not less than 70 mm [23]. Bringing the known conditions into Equation (4) can result in the maximum number of

axial straw-knife rotary planes for 3, considering that a too-small number will lead to an arrangement in the middle of the machine straw-knife-roller in which the results of the lateral cleaning of the straw cannot be thrown out, which can easily cause the blockage of the straw-clearing and straw-covering device, and the design of straw-knife-roller axial straw-knife rotary plane is 3. With Equation (3), we can obtain a number of rotary-plane circumferential straw-knives equal to 3 or 4; the design value used this paper is selected is 4. The known conditions in Equation (4) can be obtained axially adjacent with straw-knife spacing of 80 mm.

*2.2. Methods*

The straw-clearing and covering planter can sort out the seedbed without straw mulching before sowing and fertilizing, which could effectively prevent the straw from blocking the touching parts of the sowing and fertilizing components, improve the quality of sowing and fertilizing and increase the operational efficiency. At the same time, the use of straw to cover the ground surface after sowing can reduce mechanical erosion of the ground surface by rainwater and prevent soil crusting to ensure seed emergence quality [24]. Studies have shown that the selection of operating parameters significantly affects the performance of work efficiency, the straw-removal rate and power consumption. To improve the efficiency and the straw-cleaning rate of the straw-cleaning device and to reduce the power consumption of the straw-cleaning device, its working parameters need to be optimized. To avoid a large number of tedious repetitive tests, reduce labor intensity, improve work efficiency, reduce research costs and analyze the influence law of the working parameters on the evaluation index from a detailed perspective, EDEM software was used to establish a straw-clearing device–straw–soil discrete element simulation model, combined with the quadratic-regression orthogonal center-of-rotation combination test method for virtual simulation optimization test research, to obtain the optimal combination of parameters.

2.2.1. Simulation Model Construction

SolidWorks 2017 (Dassault Systèmes, Paris, France) 3D software was applied to model the straw-clearing device on an equal scale. Due to the large working width of the straw-clearing device and the symmetrical arrangement of the straw-clearing-knife rollers in the center line of the machine, the straw was thrown to both sides. Taking into account the efficiency and effectiveness of the simulation, only four groups of straw-cleaning rollers, the frame, the straw pressure plate and the straw barriers were retained in the modeling, and the parts not related to the straw-cleaning process were removed; the model was simplified [25], and the solid model assembly (X_T) format was imported into EDEM 2018 (Altair Engineering, Troy, MI, USA) software. According to the actual production materials used, the material of the straw press plate, straw barriers, and the front and rear guards of the frame model was set to Q235A, and the material of the straw-cleaning-knife roller model was set to 65 Mn.

As shown in Figure 4a, to simulate a field operation environment, a virtual soil tank model of ($L \times W$) 10,000 mm $\times$ 2600 mm was established; soil particles were simulated with a 5 mm-diameter sphere, and a soil layer with a monopoly height of 150 mm was generated using the gravity deposition method, and the vertical load required to generate the measured soil density was loaded above the soil layer to make the virtual soil layer consistent with the actual soil layer properties [26]. To accurately describe the straw transportation law, considering the diversity and complexity of maize straw morphology and combined with the actual situation in the field, a long linear model consisting of spherical particles with diameters of 16 mm, a spherical center spacing of 4 mm, and lengths of 80 mm were used as the straw particle model, and three groups of different sizes of straw were arranged on this basis as shown in Figure 4c, and a particle plant was set above the soil layer to generate 40 mm thickness and 241 kg/m$^3$ straw density of the corn stover layer [27].

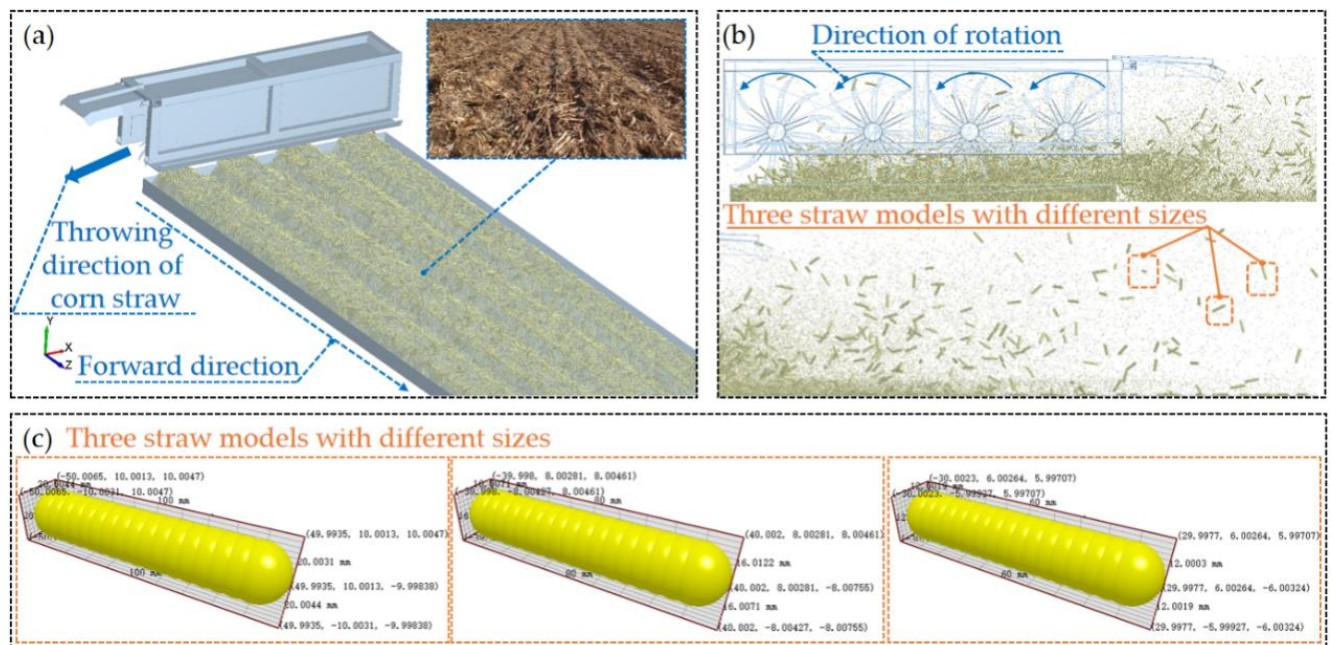

**Figure 4.** (**a**) The virtual soil tank model. (**b**) Simulation process. (**c**) The corn straw model.

To improve the quality of discrete element simulation, reasonable material contact models and correct edge parameters need to be selected. The Hertz–Mindlin model with bonding contact was set up between soil particles, and the Hert–Mindlin contact model was used between straw and straw, straw and straw-cleaning-knife roller, straw and soil, and soil and straw-cleaning-knife roller [28]. The contact and intrinsic parameters of each material were obtained by random-sampling-method measurements and searching the relevant literature [29–31] and are shown in Table 1. The straw-cleaning device model was introduced into the corn-straw-covered soil trough, and the straw–soil discrete-element-simulation model of the straw-removal device was constructed, setting the straw-removal device at one end of the trough for initial operation; the positive direction of *z*-axis was the operating direction of the device, while the negative direction of the *x*-axis was the direction of the corn straw being thrown sideways and the straw-cleaning-knife roller rotated counterclockwise around the positive direction of the *z*-axis as shown in Figure 4b. A fixed time step of $5.76 \times 10^{-5}$ s was set, which was 10% of the Rayleigh time step; the cell grid size was set to 3 times the average particle diameter; the total simulation duration was 15 s, and the simulation data were saved every 0.01 s [32]. To ensure the continuity of the simulated motion of straw particles, a single simulation was run for 20 s, and only the experimental results for 12 s within the stable working interval were extracted for subsequent statistical analysis.

**Table 1.** Material physical and contact mechanical properties parameters.

| Properties | Q235A/Source | 65Mn/Source | Straw/Source | Soil/Source |
|---|---|---|---|---|
| Density/(kg·m$^{-3}$) | 7850 /Wang et al. [29] | 7800 /Wang et al. [29] | 241 /By trial and error | 2650 /By trial and error |
| Shear's modulus/Pa | $7.9 \times 10^{10}$ /Wang et al. [29] | $7.96 \times 10^{10}$ /Wang et al. [29] | $1.0 \times 10^6$ /Wang et al. [29] | $1.0 \times 10^6$ /Wang et al. [29] |
| Poisson's ratio | 0.3 /Wang et al. [29] | 0.3 /Wang et al. [29] | 0.4 /Wang et al. [29] | 0.34 /Wang et al. [29] |
| Coefficient of rolling friction (to straw) | 0.01 /Wang et al. [29] | 0.01 /Wang et al. [29] | 0.01 /Tian et al. [30] | 0.05 /Tian et al. [30] |

**Table 1.** *Cont.*

| Properties | Q235A/Source | 65Mn/Source | Straw/Source | Soil/Source |
|---|---|---|---|---|
| Coefficient of friction (to straw) | 0.3 /By trial and error | 0.3 /By trial and error | 0.3 /By trial and error | 0.5 /By trial and error |
| Coefficient of restitution (to straw) | 0.3 /Tian et al. [30] | 0.3 /Tian et al. [30] | 0.3 /Tian et al. [30] | 0.5 /Tian et al. [30] |
| Coefficient of rolling friction (to soil) | 0.11 /Matin et al. [31] | 0.11 /Matin et al. [31] | 0.05 /By trial and error | 0.2 /By trial and error |
| Coefficient of friction (to soil) | 0.65 /By trial and error | 0.65 /By trial and error | 0.3 /By trial and error | 0.3 /By trial and error |
| Coefficient of restitution (to soil) | 0.6 /Matin et al. [31] | 0.6 /Matin et al. [31] | 0.3 /Matin et al. [31] | 0.6 /Matin et al. [31] |

### 2.2.2. Simulation Test Program

Referring to Chinese national standards DG/T 028-2019 "No-tillage planter" and GB/T 20865-2017 "No (minimum) tillage fertilizer planter", this test was implemented. The operating speed, speed of the knife roller and the penetration depth of the knife were used as the test factors, and the rate of cleaning straw was the evaluation index 1 to characterize the effect of straw-clearing and blockage-prevention, and the equivalent power consumption was the evaluation index 2 to characterize the operating economy of the unit. A three-factor, five-level quadratic-regression orthogonal center-of-rotation combination test method [33] was used to evaluate the working performance of the straw-clearing and covering planter. The test-factor levels were coded as shown in Table 2, and a total of 23 parameter combinations were tested.

**Table 2.** Test-factors coding.

| Test Factors | | Coded Value | | | | |
|---|---|---|---|---|---|---|
| | | $-1.682$ | $-1$ | 0 | 1 | 1.682 |
| $x_1$ | Operating speed $v$/(km/h) | 5.4 | 5.8 | 6.3 | 6.8 | 7.2 |
| $x_2$ | Speed of knife roller $n$/(rpm) | 400 | 441 | 500 | 559 | 600 |
| $x_3$ | Penetration depth of knife $d$/(mm) | 30 | 34 | 40 | 46 | 50 |

In the simulation tests, different combinations of parameters (operating speed, speed of the knife roller and the penetration depth of the knife) were set for the straw-removal device. Before and after each test, five randomly selected (200 mm long × 200 mm wide) straw-cleaning areas within the working width of each effective straw-cleaning roller (four rows in total) were used as collection points, and a total of 20 collection points were constructed, and the straw mass at each collection point was weighed in turn by the *Solve Report* module. In each simulation, 5 points of time were randomly selected as data-extraction points for all levels of straw-cleaning-knife rollers within the stable working interval, and a total of 20 extraction points were set up, and the torque and resistance were extracted by the Torque and Total Energy modules. The corresponding straw-removal rate and the equivalent power consumption were calculated according to Equation (5).

$$
\begin{cases}
Y_1 = \dfrac{\sum\limits_{i=1}^{4} Q_i - \sum\limits_{i=1}^{4} W_i}{\sum\limits_{i=1}^{4} Q_i} \times 100\% \\[4mm]
Y_2 = \dfrac{\sum\limits_{i=1}^{4} \sum\limits_{t=1}^{5} \left( \dfrac{M_{it} n}{9550} + \dfrac{F_{it} v}{3600} \right)}{5} \\[4mm]
Q_i = \dfrac{\sum\limits_{j=1}^{5} Q_{ij}}{5} \\[4mm]
W_i = \dfrac{\sum\limits_{j=1}^{5} W_{ij}}{5}
\end{cases}
\tag{5}
$$

where $Y_1$ is the rate of straw cleaning, %; $Y_2$ is the equivalent power consumption, kW; $Q_i$ is the amount of corn straw cover before the i-stage straw-removal-knife roller corresponding to the measurement area operation, kg; $W_i$ is the amount of corn straw cover after the *i*-stage straw-cleaning-knife roller corresponding to the measurement area operation, kg; $Q_{ij}$, is the jth sample of $Q_i$, kg; $W_{ij}$ is the jth sample of $Q_i$, kg; $M_{it}$ is the torque corresponding to the *t*th time point of the *i*-stage straw-cleaning-knife roller, N·m; $F_{it}$ is the resistance corresponding to the *t*th time point of the *i*-stage straw-cleaning-knife roller, N.

2.2.3. Field Test Program

To verify the correctness of the discrete element simulation, a field experiment was carried out on the combination of optimal parameters of the discrete element simulation test. As shown in Figure 5a, the field test was conducted 22–27 April 2019, at the air force farm in Keshan County, Heilongjiang Province, on a plot of raw corn stubble after crushing with a Foton Lovol Gushen harvester (Foton Lovol, Weifang, China). The previous crop was the Danuo 6 maize variety; the average length of the corn stover was 80 mm; the water content was 26.2%; the mulch amount was 1.57 kg/m$^2$, and the average stubble height was 210 mm. The soil type was black clay; the average soil hardness was 17.3 kg/cm$^2$ at depth 0–10 cm, and the average soil water content was 20.6%. The test instruments and equipment included a Valmet 2104 tractor (Valmet, Espoo, Finland), a 2BMFJ-BL8 straw-clearing and covering planter, a SZ-3 soil-hardness meter, a SU-LB soil-moisture meter, an ACS-30 electronic scale, a JM5937A real-time data acquisition device, a JNNT-0 torque-measuring sensor, a SFZ001 tensile-force-measuring sensor, a homemade rotational-speed-measuring instrument, a digital camera, a meter ruler, a stopwatch, etc. The instrumentation connections are shown in Figure 5b.

Field validation tests were conducted under three operating conditions, with operating speeds of 5.5, 5.8 and 6.2 km/h, a straw-cleaning-roller speed of 500 rpm and a straw-cleaning knife-penetration depth of 40 mm. The straw-cleaning-rate data was measured according to Section 2.2.2, and each test was repeated 5 times, and the average value was taken. The equivalent power consumption was measured by the tension sensor fixed between the tractor and the test device to measure the traction force, the torque sensor between the tractor and the drive shaft to measure the straw-clearing torque, and through the real-time data-acquisition device for data collection and storage, each group of tests was repeated 5 times, and the average value was taken, and the formula was calculated as follows:

$$
y_2 = \frac{\dfrac{T_m n_m}{9950} + \dfrac{F_m v_m}{3600}}{8}
\tag{6}
$$

where $T_m$ is the measurement of the average torque, N·m; $n_m$ is the tractor power-output-shaft average speed, rpm; $F_m$ is the measurement of the average tractive force, N; $v_m$ is the average speed of the operation of the implement, km/h.

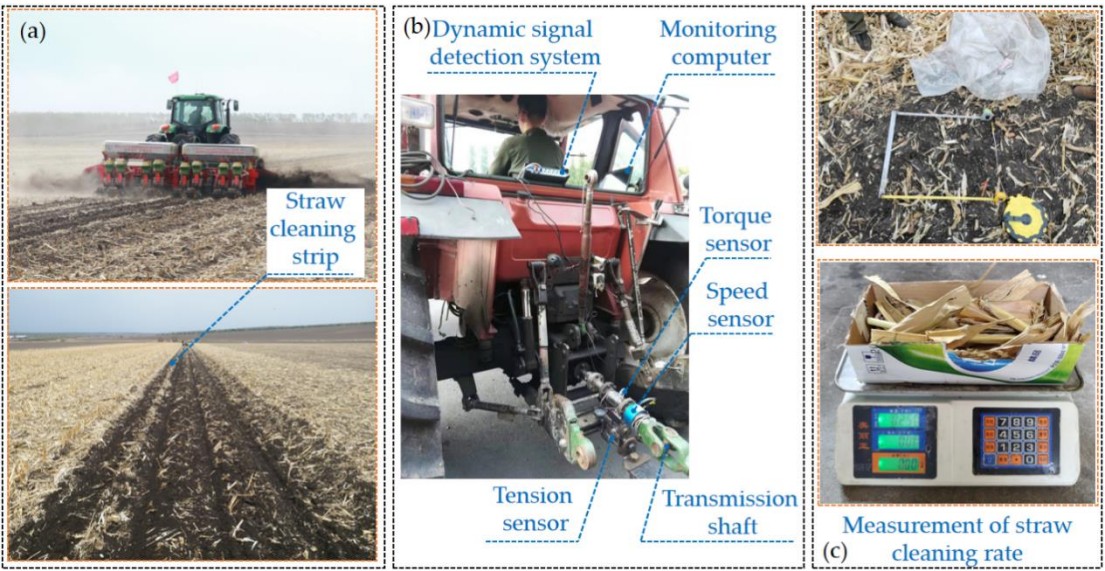

**Figure 5.** Equipment connection and evaluation index measurement. (**a**) Test environment and process; (**b**) connection of the measuring equipment; (**c**) the measurement of the straw-cleaning rate.

## 3. Results and Discussion

According to the design scheme provided by Design-Expert 8.0.6 (Stat-Ease, Minneapolis, MN, USA), the simulation test is completed, and the results are shown in Table 3, including 14 analysis factors and 9 null tests of the estimation errors. Design-Expert 8.0.6 was applied to perform ANOVA on the test results, and the results are shown in Table 4. The F-test was conducted at a confidence level of 0.05, and the regression model of evaluation indices was obtained after excluding the insignificant term as follows:

$$\begin{cases} Y_1 = 89.91 - 0.71X_1 + 0.36X_2 + 0.29X_3 + 0.46X_1X_2 - 0.39X_1X_3 - 0.33X_1^2 - 0.95X_2^2 - 1.22X_3^2 \\ Y_2 = 7.98 + 0.13X_1 + 0.18X_2 + 0.74X_3 - 0.19X_1X_3 + 0.34X_2^2 + 0.11X_3^2 \end{cases} \quad (7)$$

**Table 3.** Test schemes and results.

| Test Number | Factors | | | Evaluation Index | |
|---|---|---|---|---|---|
| | $X_1$ | $X_2$ | $X_3$ | $Y_1$/% | $Y_2$/kW |
| 1 | −1 | −1 | −1 | 87.17 | 7.46 |
| 2 | 1 | −1 | −1 | 84.64 | 7.88 |
| 3 | −1 | 1 | −1 | 87.55 | 7.49 |
| 4 | 1 | 1 | −1 | 87.89 | 8.33 |
| 5 | −1 | −1 | 1 | 89.03 | 9.12 |
| 6 | 1 | −1 | 1 | 85.98 | 8.69 |
| 7 | −1 | 1 | 1 | 88.29 | 9.45 |
| 8 | 1 | 1 | 1 | 87.06 | 9.62 |
| 9 | −1.682 | 0 | 0 | 90.38 | 8.03 |
| 10 | 1.682 | 0 | 0 | 87.84 | 8.47 |
| 11 | 0 | −1.682 | 0 | 86.77 | 8.72 |
| 12 | 0 | 1.682 | 0 | 87.95 | 9.16 |
| 13 | 0 | 0 | −1.682 | 86.04 | 6.98 |
| 14 | 0 | 0 | 1.682 | 87.12 | 9.62 |
| 15 | 0 | 0 | 0 | 89.47 | 8.01 |
| 16 | 0 | 0 | 0 | 90.27 | 7.68 |
| 17 | 0 | 0 | 0 | 90.16 | 8.21 |
| 18 | 0 | 0 | 0 | 90.77 | 8.16 |
| 19 | 0 | 0 | 0 | 89.63 | 7.69 |
| 20 | 0 | 0 | 0 | 89.98 | 7.84 |
| 21 | 0 | 0 | 0 | 89.77 | 8.36 |
| 22 | 0 | 0 | 0 | 89.73 | 8.01 |
| 23 | 0 | 0 | 0 | 89.36 | 7.84 |

**Table 4.** ANOVA.

| Source of Variance | $Y_1$/% | | | | | $Y_2$/kW | | | | |
| --- | --- | --- | --- | --- | --- | --- | --- | --- | --- | --- |
| | Sum of Squares | Freedom | Mean Square | F | p | Sum of Squares | Freedom | Mean Square | F | p |
| Model | 56.97 | 9 | 6.33 | 24.01 | <0.0001 ** | 12.47 | 9 | 1.39 | 71.68 | <0.0001 ** |
| $X_1$ | 7.21 | 1 | 7.21 | 27.36 | 0.0002 ** | 0.21 | 1 | 0.21 | 10.64 | 0.0062 ** |
| $X_2$ | 1.46 | 1 | 1.46 | 5.53 | 0.0352 * | 0.22 | 1 | 0.22 | 11.61 | 0.0047 ** |
| $X_3$ | 1.58 | 1 | 1.58 | 5.98 | 0.0295 * | 11.04 | 1 | 11.04 | 571.19 | <0.0001 ** |
| $X_1 X_2$ | 1.39 | 1 | 1.39 | 5.26 | 0.0392 * | 0.0007 | 1 | 0.0007 | 2.032 | 0.0861 |
| $X_1 X_3$ | 1.49 | 1 | 1.49 | 5.64 | 0.0336 * | 0.11 | 1 | 0.11 | 5.59 | 0.0343 * |
| $X_2 X_3$ | 0.33 | 1 | 0.33 | 1.26 | 0.2820 | 0.03 | 1 | 0.03 | 1.55 | 0.2347 |
| $X_1^2$ | 1.94 | 1 | 1.94 | 7.35 | 0.0178 * | 0.021 | 1 | 0.021 | 1.08 | 0.3182 |
| $X_2^2$ | 16.97 | 1 | 16.97 | 64.36 | <0.0001 ** | 0.037 | 1 | 0.037 | 1.94 | 0.1870 |
| $X_3^2$ | 25.07 | 1 | 25.07 | 95.10 | <0.0001 ** | 0.81 | 1 | 0.81 | 41.75 | <0.0001 ** |
| Residual | 3.43 | 13 | 0.26 | | | 0.25 | 13 | 0.019 | | |
| Lack off it | 1.33 | 5 | 0.27 | 1.01 | 0.4687 | 0.11 | 5 | 0.022 | 1.22 | 0.3831 |
| Pure Error | 2.10 | 8 | 0.26 | | | 0.14 | 8 | 0.018 | | |
| Corrected Total | 60.40 | 22 | | | | 12.72 | 22 | | | |

Note: ** indicates highly significant ($p < 0.01$); * indicates significant ($0.01 < p < 0.05$).

### 3.1. ANOVA

The regression coefficients in the regression models of each evaluation index were analyzed by ANOVA, and according to the loss-of-fit values of the regression models of evaluation indices in Table 4, it can be seen that $P_{L1} = 0.4678 > 0.05$ and $P_{L2} = 0.3226 > 0.05$ (both not significant), indicating that there is no loss factor in the regression analysis and that the regression models fit well. The model values of the regression model $P_{M1} < 0.0001$ and $P_{M2} < 0.0001$ (both highly significant) indicate that the regression results have some reliability.

Through ANOVA, it can be seen that the operating speed has a highly significant effect on the rate of straw cleaning, that the speed of the knife roller and the penetration depth of the knife have a significant effect on the rate of straw cleaning and that the effects are, in descending order, operating speed, the penetration depth of the knife, the speed of the knife roller, the interaction between operating speed and the speed of the knife roller and the interaction between operating speed and the penetration depth of the knife have a significant effect on the rate of straw cleaning. All of the test factors have a significant effect on the equivalent power consumption; in descending order of influence, the penetration depth of the knife, the speed of the knife roller, operating speed and the interaction between operating speed and the penetration depth of the knife have a significant effect on the equivalent power.

### 3.2. Parameter Combination Optimization

Our team found that the actual no- and minimum-tillage seeding quality met the agronomic requirements when the straw removal rate reached 90% or more [34]. The optimization principle was developed by considering the agronomic requirements and the simulation limitations to reduce the equivalent power consumption under the premise of ensuring the straw-cleaning and anti-blocking performance of the straw-cleaning device. A multi-objective variable optimization method was used to optimize the straw-removal device with each factor level interval as the constraint. Since the speed of the knife roller is easier to control in the actual operation process, a 500 rpm speed of the knife roller was selected as the condition for optimization. The result is shown in Figure 6a; the yellow area in the figure is the best working area, considering that the decrease in the penetration depth of the knife will cause a decrease in the corn-stubble-cleaning rate, so a penetration depth of the knife of 40 mm is selected, i.e., when the optimized parameter combination is an 5.5–6.2 km/h operation speed, a 500 rpm speed of the knife roller, a penetration depth of the knife of 40 mm, a straw-cleaning rate of more than 90% and an equivalent power consumption of less than 8 kW. The corresponding parameters are shown in Figure 6b.

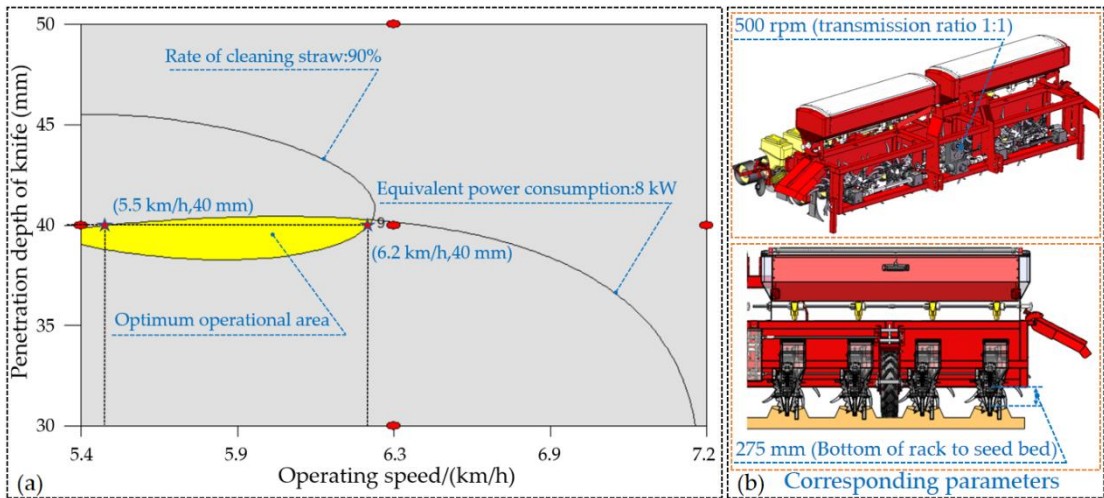

**Figure 6.** (**a**) Optimization result. (**b**) Corresponding parameters are required for the field test.

### 3.3. Field Test

To verify the accuracy of the discrete element simulation tests, field-performance verification tests were conducted on the designed straw-clearing and covering planter based on the optimized parameter combinations, and the test results are shown in Table 5.

**Table 5.** Results and comparison of the field validation test.

| Properties | Operating Speed *v*/(km/h) | | |
|---|---|---|---|
| | **5.4** | **5.8** | **6.2** |
| Predicted value of the straw-clearing rate/% | 93.28 | 91.66 | 90.79 |
| Test value of the straw-clearing rate/% | 95.76 | 93.14 | 92.43 |
| Relative error/% | 2.66 | 1.61 | 1.81 |
| Predicted value of the equivalent power consumption/kW | 7.54 | 7.65 | 7.76 |
| Test value of the equivalent power consumption/kW | 7.89 | 7.94 | 7.98 |
| Relative error/% | 4.64 | 3.79 | 2.84 |

From the test data obtained, it can be seen that the straw-clearing and covering planter with the optimized combination of operating parameters shows a better working performance. The test results on the straw-cleaning rate are consistent with the simulation results, and the relative error between the experimental results and the simulation results of equivalent power consumption error is large but within the acceptable range. The cause of the error may be that the simulation environment is too ideal and the root stubble is not taken into account, while the water content and the bulk density of the soil in the field vary greatly, increasing the resistance to operation. The field test results show that the established discrete element simulation model and the virtual simulation optimization test have certain accuracy and validity and that the optimized parameter combinations are credible.

### 3.4. The Impact of Each Factor on the Performance Evaluation Index

To express the influence of each factor on the evaluation indices, the quadratic-regression equations of the two evaluation indices mentioned above were dimensionally reduced. When the penetration depth of the knife is at the design center (40 mm), the effect of operating speed and the speed of the knife roller on the rate of straw-cleaning is shown in Figure 7a. When the operating speed is certain, the straw-clearing rate tends to increase first and then decrease with the increase of the straw-clearing-knife-roller speed,

which is mainly because, when the straw speed of the knife roller is low, the initial speed of the straw thrown by the straw-clearing knife is low and the time required to transport the straw to the next level of the straw-clearing-knife-roller straw-clearing belt increases, causing the straw to stay longer inside the straw-clearing device, resulting in a lower straw-clearing rate, as shown in Figure 8a. When the straw-clearing-knife-roller speed is high, some straw is brought back by the straw-clearing knife to the first level of the straw-clearing-knife-roller-clearing straw belt, another part of the straw throwing-height is higher, and the straw-clearing-knife roller and guard plate collision after the disorderly movement increased the straw's residence time in the straw-clearing device, resulting in the straw-clearing rate being reduced as shown in Figure 8b. The simulated motion pattern of the corn-straw model is consistent with the actual motion pattern of the corn straw under high-speed photography [35]. In the straw-cleaning-knife-roller speed range of 400–500 rpm, the straw-cleaning rate decreases with the increase in operating speed, which is mainly because, with the increase in operating speed, the straw-feeding amount per unit time of the straw-cleaning device increases, requiring the straw to be thrown out by the straw-cleaning-knife roller for less time, but some of the straw is not thrown out in time, leading to the decrease in the straw-cleaning rate, as shown in Figure 7c. With a straw-cleaning-knife-roller speed in the 500–600 rpm range, the straw-cleaning rate with the increase in operating speed first increased and then decreased, but the changing trend is not obvious, mainly because, when the operating speed of the lower straw-cleaning device's internal straw flow is low, it is vulnerable to the above-described disorderly movement of straw, resulting in a low rate of straw cleaning; with the increase in the operating speed of the straw-cleaning anti-blocking device, the amount of straw it can transport increased, and the stable flow of straw transport reduces the impact of the disorderly movement of straw and improves the rate of straw cleaning, as shown in Figure 8d. The stable straw-transport flow also helps the straw to be spread evenly and thus contributes to increasing the covering uniformity of straw [36]. When the operating speed is higher, the straw discharge of the straw-clearing device reaches saturation, which is the same reason why the straw-clearing rate varies with the operating speed in the range of 400–500 rpm of the straw-clearing-knife-roller speed. Further analysis shows that the surface response of the straw-clearing rate changes faster in the direction of the operating speed than in the direction of the straw-clearing-knife-roller speed, indicating that the operating speed has a more significant effect on the straw-clearing rate than the straw-clearing-knife-roller speed.

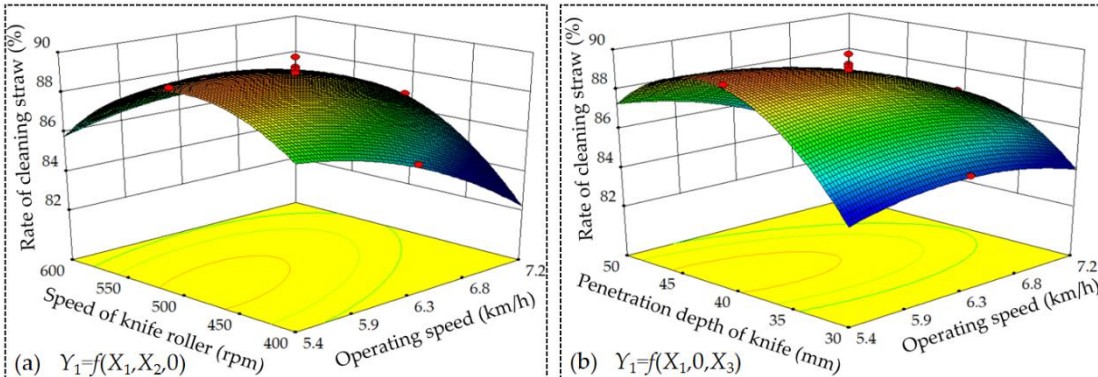

**Figure 7.** (**a**) Effect of interaction between operating speed and speed of knife roller on rate of straw cleaning. (**b**) Effect of interaction between operating speed and penetration depth of knife on rate of straw cleaning.

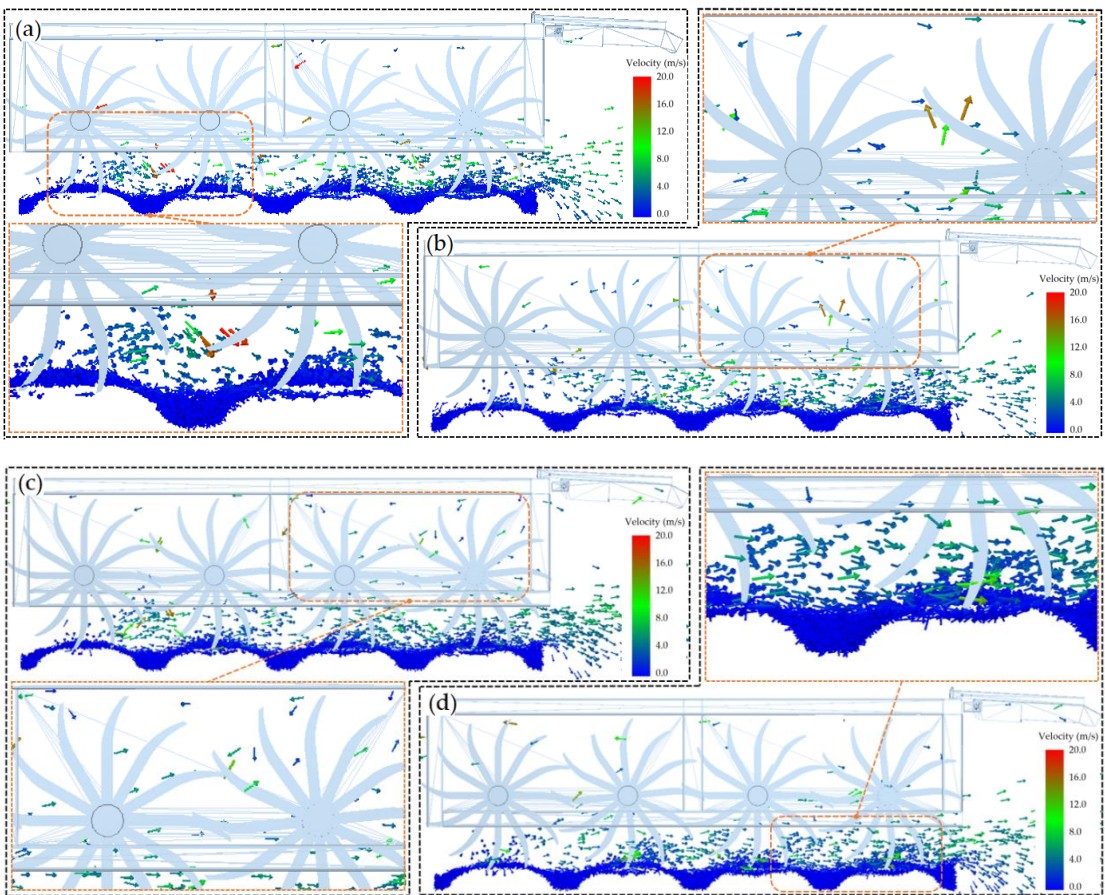

**Figure 8.** EDEM simulation diagram (**a**). Simulation diagram when *v* = 6.3 km/h, *n* = 400 rpm, *d* = 40 mm (**b**). Simulation diagram when *v* = 6.3 km/h, *n* = 600 rpm, *d* = 40 mm; (**c**). Simulation diagram when *v* = 5.4 km/h, *n* = 500 rpm, *d* = 40 mm (**d**). Simulation diagram when *v* = 7.2 km/h, *n* = 500 rpm, *d* = 40 mm.

When the speed of the knife roller is at the design center point (500 rpm), the effect of operating speed and the penetration depth of the knife on the straw-cleaning rate is shown in Figure 7b. When the operating speed is certain, the straw-cleaning rate tends to increase and then decrease with the increase of the depth of penetration of the knife, which is mainly because, when the depth of penetration of the straw-cleaning knife into the soil depth is low and the width of the straw-cleaning belt is small, the straw to be cleaned in the straw-cleaning belt is easily blocked by the straw outside the straw-cleaning belt, causing part of the straw to remain in the straw-cleaning belt or not be thrown out in time, resulting in a low rate of straw cleaning, as shown in Figure 9a. As the penetration depth of the knife increases, the width of the straw-clearing belt increases, leading to a reduction in the amount of straw outside the straw-clearing belt, thus reducing the impact of treating the clean straw transport and improving the straw-clearing rate, but when the depth of penetration of the straw-clearing knife into the soil depth is deep, the amount of moved soil makes it easy easy to form soil–straw bonds, as shown in Figure 9b, affecting the straw-clearing-knife-roller clearing-straw effect, and, in serious cases, the soil–straw accumulation on the rotating knife shaft can not work properly, leading to a reduction in the straw-clearing rate. When the straw-cleaning knife penetrates the soil at a depth of 30–40 mm, the rate of straw cleaning, with the increase in operating speed, is first increased and then decreased, but the trend is not obvious, mainly because when the operating speed is low, the larger vibration inertia of the straw-cleaning-knife roller will cause disturbance to the soil and straw on both sides of the outside of the straw-cleaning belt, causing the straw outside the straw-cleaning belt to flow back into the straw-cleaning belt, resulting in a low

rate of straw cleaning; with the increase in operating speed, the frequency of disturbance to the soil and straw by the straw-cleaning-knife roller is reduced and the flow of straw inside the straw-cleaning anti-blocking device increases, reducing the phenomenon of straw flowing back into the straw-cleaning belt and increasing the rate of straw cleaning. When the operating speed is high, the straw-clearing knife moves forward to push the soil and straw obviously leads to too much straw extrusion in the pressure straw plate below, affecting the straw-clearing device passability and the straw-clearing-knife-roller clearing-straw effect, resulting in the straw-clearing rate being reduced, as shown in Figure 9c. The stacking pattern of the straw model was very similar to the results of the field test [37]. When the straw-cleaning knife penetrates into the soil depth in the 40–50 mm range, the rate of straw cleaning with the increase of operating speed shows a decreasing trend, and the changing trend is very obvious, which is mainly because, with the increase of operating speed, the straw-cleaning knife's soil extrusion causes a serious part of the straw to not be thrown directly into the soil or to not be bonded with the soil, intensifying the reduction of the rate of straw cleaning, as shown in Figure 9d. A further comparison of the operating speed and the penetration depth of the straw-cleaning knife into the soil depth with the rate of straw-cleaning shows a significant interaction.

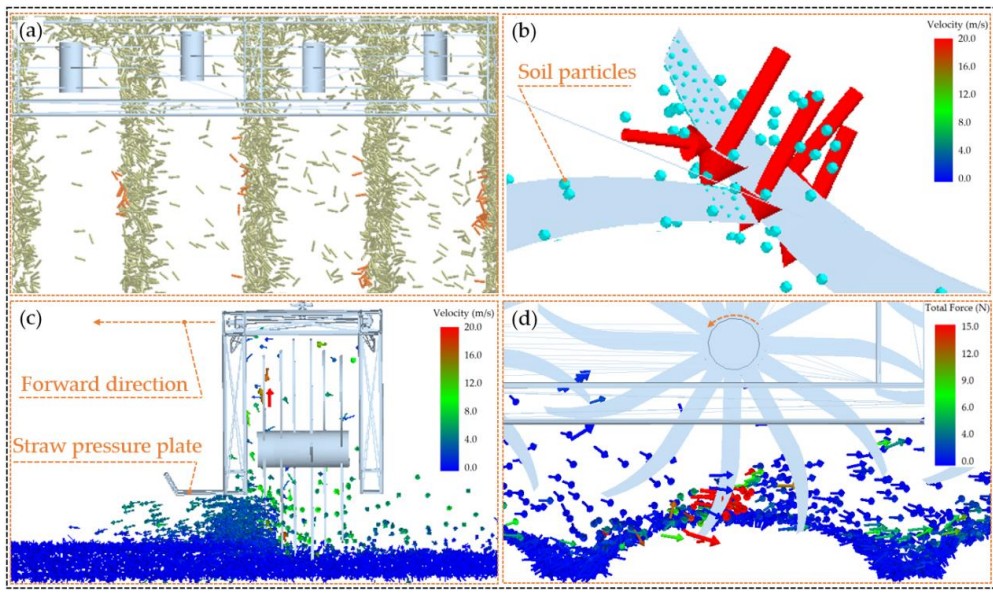

**Figure 9.** EDEM simulation diagram. (**a**) Simulation diagram when *v* = 6.3 km/h, *n* = 500 rpm, *d* = 30 mm. (**b**) Simulation diagram when *v* = 6.3 km/h, *n* = 500 rpm, *d* = 50 mm. (**c**) Simulation diagram when *v* = 5.4 km/h, *n* = 500 rpm, *d* = 40 mm. (**d**) Simulation diagram when *v* = 7.2 km/h, *n* = 500 rpm, *d* = 40 mm.

When the speed of the knife roller is at the design center point (500 rpm), the effect of operating speed and the penetration depth of the knife on the equivalent power consumption is shown in Figure 10a. When the operation speed is certain, the equivalent power consumption increases with the increase of the penetration depth of knife, which is mainly because, with the increase of the depth of penetration of the straw-clearing knife into the soil, the straw width increases and the quality of straw cleared by the straw cleaning device increases while the amount of moved soil increases, which increases the equivalent power consumption. When the penetration depth of the knife is certain, the equivalent power consumption with the operating speed increases, with the increasing trend, on the one hand, in the case of constant resistance with the increase in operating speed leading to increased power consumption; on the other hand, in the case of a constant speed of the straw-cleaning-knife roller operating speed, it increases the proportion of congestion and increased resistance, resulting in increased equivalent power consumption. The surface

response shows that, when the depth of penetration of the straw-cleaning knife into the soil is 30 mm, the equivalent power consumption tends to increase with the increase of the penetration depth of the straw-removal knife, and the change is not obvious, but when the penetration depth of the straw-cleaning knife into the soil is 50 mm, the changing trend is very obvious, indicating that the penetration depth of the knife has a more significant effect on the equivalent power consumption than the operating speed. The effect of the penetration depth of the knife on the equivalent power consumption is shown in Figure 11a. When the operating speed is at the design center point (6.3 km), the influence of the speed of the knife roller on the equivalent power consumption is shown in Figure 10b. The equivalent power consumption tends to decrease and then increase with the increase of the speed of the knife roller, mainly because, when the straw-cleaning-knife-roller speed is low, the straw-cleaning-knife throwing performance is reduced, resulting in congestion in the front of the straw-cleaning device, resulting in high equivalent power consumption. The rotation plane of the straw-cleaning roller is perpendicular to the operating speed, which makes this phenomenon more obvious [38]. With a knife-roller speed of 400–500 rpm, as the straw-cleaning-knife-roller speed increases, the straw- and soil-throwing performance improves and the equivalent power consumption decreases. With a knife-roller speed of 500–600 rpm, the equivalent power consumption gradually increases with the increase of straw-cleaning-knife-roller speed, and the rate of increase gradually increases. The effect of straw cleaning knife shaft speed on the equivalent power consumption is shown in Figure 11b.

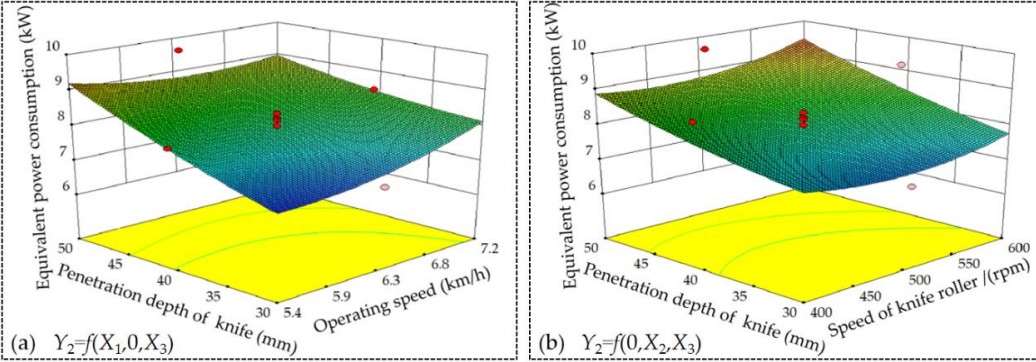

**Figure 10.** (**a**) Effect of interaction between operating speed and penetration depth of knife on equivalent power consumption. (**b**) Effect of interaction between speed of knife roller and penetration depth of knife on equivalent power consumption.

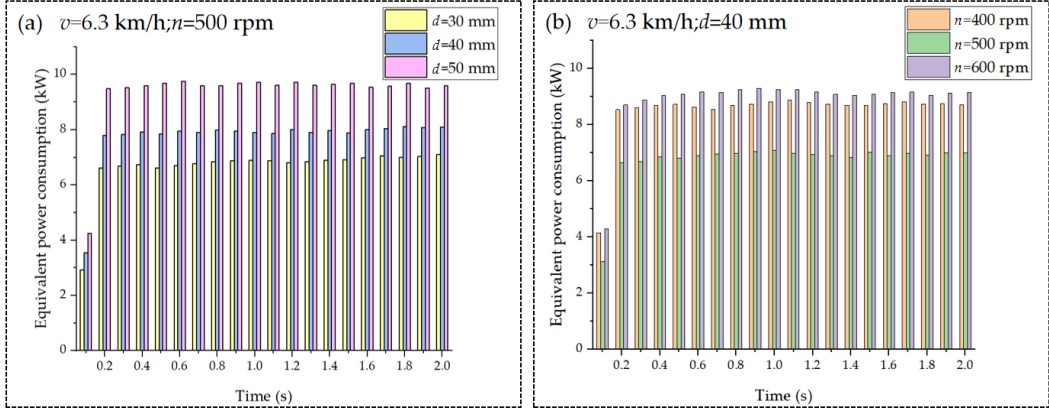

**Figure 11.** Equivalent power consumption change graph. (**a**) Diagram of the effect of the penetration depth of the knife on equivalent power consumption. (**b**) Diagram of the effect of the speed of the knife roller on equivalent power consumption.

## 4. Conclusions

A clearing and covering minimum-tillage planter for corn stubble was developed for the large-scale planting pattern in northeast China, which realized the high quality and efficient minimum-tillage sowing operation under the condition of the full coverage of corn straw. The overall structure and working principle of the straw-clearing and covering planter are described, and the straw-clearing device is emphatically designed, of which the structure and working parameters have been determined. EDEM 2018 (Altair Engineering, Troy, MI, USA) was used to establish the straw-removal device–straw–soil discrete element simulation model, combined with the three-factor five-level quadratic-regression orthogonal center-of-rotation combination test method, to determine the optimal combination of parameters affecting the working performance of the straw-cleaning device, when the operating speed is 5.5–6.2 km/h, the speed of the knife roller is 500 rpm, the penetration depth of the knife is 40 mm, the rate of straw-cleaning is more than 90%, and the equivalent power consumption is less than 8 kW. The field validation tests were implemented for the optimized parameter combinations, and the results of each test matched the predicted values with a relative error less than or equal to 4.64%, and the test results proved that the established discrete element simulation model and virtual simulation optimization tests have certain accuracy and effectiveness.

**Author Contributions:** Conceptualization, S.H., S.W. and Z.J.; methodology, S.H., S.W. and Z.J.; software, S.H., S.W. and Z.J.; validation, S.H. and X.Z.; writing—original draft preparation, S.H; writing—review and editing, S.W. and X.Z.; project administration, S.H. All authors have read and agreed to the published version of the manuscript.

**Funding:** This research was funded by the National Natural Science Foundation of China (Grant No. 32101628) and the Natural Science Foundation of Heilongjiang Province (Grant No. LH2021E004).

**Institutional Review Board Statement:** Not applicable.

**Informed Consent Statement:** Not applicable.

**Data Availability Statement:** Not applicable.

**Conflicts of Interest:** The authors declare no conflict of interest.

## Nomenclature

| | | |
|---|---|---|
| $R$ | Knife-roller turning radius | mm |
| $d$ | Penetration depth of the straw-cleaning knife | mm |
| $w$ | Width of the straw-cleaning belt | mm |
| $D$ | Diameter of the knife shaft | mm |
| $l$ | Monopoly distance | mm |
| $s_{\max}$ | Maximum length of the straw | mm |
| $\theta_{\min}$ | Minimum stubble-clearing angle | rad |
| $\theta_{\max}$ | Maximum stubble-clearing angle | rad |
| $n_1$ | Number of circuit-erential straw-clearing knives in the same rotary plane | |
| $n_2$ | Number of axial straw-clearing knife rotary planes | |
| $l_0$ | Axial adjacent straw-knife spacing | mm |
| $v$ | Operating speed | km/h |
| $n$ | Speed of the knife roller | rpm |
| $Y_1$ | Rate of straw-cleaning | % |
| $Y_2$ | Equivalent power consumption | kW |
| $Q_i$ | Amount of corn straw cover before the i-stage straw-removal-knife roller corresponding to the measurement-area operation | kg |
| $W_i$ | Amount of corn straw cover after the *i*-stage straw-cleaning-knife roller corresponding to the measurement-area operation | kg |

| $Q_{ij}$ | The *j*th sample of $Q_i$ | kg |
| $W_{ij}$ | The *j*th sample of $W_i$ | kg |
| $M_{it}$ | Torque corresponding to the *t*th time-point of the *i*-stage straw-cleaning-knife roller | N·m |
| $F_{it}$ | Resistance corresponding to the *t*th time-point of the *i*-stage straw-cleaning-knife roller | N |
| $T_m$ | Measurement of the average torque | N·m |
| $n_m$ | Tractor power-output-shaft average speed | rpm |
| $F_m$ | Measurement of the average tractive force | N |
| $v_m$ | Average speed of the operation of the implement | km/h |

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
