# Peer review of "Design and Test of the Clearing and Covering of a Minimum-Tillage Planter for Corn Stubble"

_agriculture, doi:10.3390/agriculture12081209_

Round 1

Reviewer 1 Report

1.      In the abstract, the full definition is not provided for the first use of EDEM.

2.      The references of properties in Table 1 are not provided.

3.      Normally the coded values for minimum and maximum are -2 and 2. Why are the coded values in Table 2 are -1.682 and 1.682?

4.      How did the authors decide the range of value for test factors? Based on any references?

5.      Does the design scheme of table 3 generated by a design expert? Please mention this in the manuscript.

6.      How many samples were done for tests number 1-14?

7.      Why do the repeated null test results vary for a similar setting of test factors?

8.      Please check the subscript error of the source of variance for X1X2 and remove the line after X2-square in Table 4.

9.      Please fix the label of speed in Table 5. 

Author Response

Dear Reviewer,

Thank you very much for reviewing the article " Design and Test of Clearing and Covering of Minimum-Tillage Planter for Corn Stubble " in your busy schedule and making valuable comments. The authors have carefully studied your revisions and made the following revisions and replies to the article in accordance with the relevant comments.

  1. In the abstract, the full definition is not provided for the first use of EDEM.

A: Based on the issues raised by the reviewer, I have added “discrete element method software (EDEM)” in the abstract section to provide the definition for EDEM.

  1. The references of properties in Table 1 are not provided.

A: I have added the reference sources for each parameter in Table 1.

  1. Normally the coded values for minimum and maximum are -2 and 2. Why are the coded values in Table 2 are -1.682 and 1.682?

A: The experimental factor coding values were determined according to the experimental design method used. This manuscript uses a three-factor, five-level quadratic orthogonal rotational regression test design in which the maximum and minimum test factor coding values should be distributed on a circle of radius γ. γ is calculated based on the rotational condition equation(γ=2m/4), where m is the number of test factors(3), and γ = 1.682.

  1. How did the authors decide the range of value for test factors? Based on any references?

A: Through the team's preliminary experimental research, it was found that the operating speed of the machine was negatively correlated with the sowing quality of the sowing unit, and when the operating speed was greater than 7.2 km/h, the qualified rate of sowing depth and the qualified rate of sowing grain spacing would drop significantly. To meet the agronomic requirements, the upper limit of the designed operating speed is 7.2 km/h, while the lower limit of the designed operating speed is 5.4 km/h to ensure maximum efficiency. During the pre-test study, it is founded that the root stubble removal rate can reach more than 90% when the straw cleaning knife enters the soil to a depth of 50 mm, to reduce the amount of soil disturbance, reduce power consumption and improve efficiency, the design depth of soil penetration is 30-50 mm. Considering the large diameter of corn root stubble and many roots in northeast China, the stubble cleaning effect is not ideal when the knife roller speed is lower than 400 rpm, and it is difficult to achieve a relatively clean area. Because the knife roller is driven by chain, when the knife roller speed is higher than 600rpm, the machine will produce violent vibration and noise.

  1. Does the design scheme of table 3 generated by a design expert? Please mention this in the manuscript.

A: The design scheme in Table 3 is produced by the Design-Expert 8.0.6. According to the suggestions of the reviewer, I have added “According to the design scheme provided by Design-Expert 8.0.6, the simulation test is completed, and the results are shown in Table 3” at the beginning of the third chapter (Results and Discussion) of this manuscript.

  1. How many samples were done for tests number 1-14?

A: In tests 1-14, each test was repeated three times to take the mean.

  1. Why do the repeated null test results vary for a similar setting of test factors?

A: In simulation tests 15-23, the authors regenerated a new straw layer with EDEM before the start of each simulation trial in order to better simulate the field trial environment. The straw in the new straw layer is randomly generated and varies somewhat, leading to different simulated trial results under the same operating parameters.

  1. Please check the subscript error of the source of variance for X1X2 and remove the line after X2-square in Table 4.

A: I have fixed the errors and removed the line after X2-square in Table 4.

  1. Please fix the label of speed in Table 5.

A: I have changed the Chinese in Table 5 into English

Reviewer 2 Report

I have not done a detailed analysis of the article yet. So if you have other review options, then use them. I can make a few first remarks that I think could improve this work: - the topic of the work is relevant and in general the authors have done quite a lot of research work - keywords - almost all are very long (these are sentences, not words) --formulas 1-4 look a little primitive (are they needed at all?) for figures 9 and 10 there is no color scale scale - the second paragraph of the conclusions, it seems to me, should be moved to the discussion (does not contain the necessary information for the conclusions) - references are mainly from Chinese authors, but for an international magazine on a generally ubiquitous topic, I would like to see more European and American authors)

Author Response

Dear Reviewer,

Thank you very much for reviewing the article " Design and Test of Clearing and Covering of Minimum-Tillage Planter for Corn Stubble " in your busy schedule and making valuable comments. The authors have carefully studied your revisions and made the following revisions and replies to the article in accordance with the relevant comments.

  1. The topic of the work is relevant and in general the authors have done quite a lot of research work - keywords - almost all are very long (these are sentences, not words).

A: I have revised the keywords in this manuscript (deleted the “virtual simulation experiment” and added “design, test”).

  1. Formulas 1-4 look a little primitive (are they needed at all)?

A: The key component of this manuscript design is the straw cleaning knife roller on the straw cleaning device. The theoretical source of its main parameters design is determined by the actual field environment, agronomic requirements, and mechanical processing requirements, and Equations 1-4 better reflect the author's design ideas and theoretical sources.

  1. Figures 9 and 10 there is no color scale .

A: The errors in the figures have been corrected.

  1. The second paragraph of the conclusions, it seems to me, should be moved to the discussion (does not contain the necessary information for the conclusions).

A: At the suggestion of the reviewer, I have changed the title of chapter 4 of this manuscript from " 4. Conclusion " to "4. Conclusion and discussion " and added two subsections, entitled "4.1. Conclusion" and "4.2. Discussion", and finally moved the “The research results of this paper have the following limitations…..” in "4.2. Discussion ".

  1. References are mainly from Chinese authors, but for an international magazine on a generally ubiquitous topic, I would like to see more European and American authors).

A: Based on the issues raised by the reviewer, I have updated the references in terms of conservation tillage technology, EDEM modeling, and straw parameter calibration.

Reviewer 3 Report

Some specific comments:

1. The rotation plane of the straw cleaning knife roller is perpendicular to the moving direction of the machine. How about the force of the knife roller when working?

2. The corn straw is cleaned by the straw cleaning knife roller, whether the straw being cleaned will affect the uncleaned straw during the transportation process.

3. In line 48, what is the operating speed and width of the existing no-tillage seeders. Can you give some specific data?

4. In lines 139 and 170, the title “2.1.1. Structure composition and working principle” is repeated, please check.

5. Please check the section number. Line 325 should be Section 2.2.2.

6. Please check Equation (6), the author should be more careful.

7. Table 5 contains Chinese.

8. Figure 7 is confusing. The upper and lower figures seem to be the same. Where are figures (c) and (d)? Figure 7 looks messy and should be rearranged.

Author Response

Dear Reviewer,

Thank you very much for reviewing the article " Design and Test of Clearing and Covering of Minimum-Tillage Planter for Corn Stubble " in your busy schedule and making valuable comments. The authors have carefully studied your revisions and made the following revisions and replies to the article in accordance with the relevant comments.

  1. The rotation plane of the straw cleaning knife roller is perpendicular to the moving direction of the machine. How about the force of the knife roller when working?

A: The machine designed in this manuscript is a minimum-tillage planter, the straw clearing knife into the soil shallow, and each group of straw clearing knife rollers only 1 to 2 straw clearing knife contact with the soil at any given time, the knife rollers by the horizontal direction of resistance is much smaller than the rotary resistance.

  1. The corn straw is cleaned by the straw cleaning knife roller, whether the straw being cleaned will affect the uncleaned straw during the transportation process.

A: As the machine works, the straw is transported from the middle of the machine to both sides and then thrown out sideways. The process is repeated continuously as the machine advances, and the straw being cleaned will not have an impact on the uncleared straw.

  1. In line 48, what is the operating speed and width of the existing no-tillage seeders. Can you give some specific data?

A: Most of the existing commonly used wide no-tillage seeders are six-row machines, whose operating width is generally around 4500 mm, and whose operating speed is generally around 5.4 km/h.

  1. In lines 139 and 170, the title “2.1.1. Structure composition and working principle” is repeated, please check.

A: In line 170, I have replaced “2.1.1. Structure composition and working principle” with “2.1.2. Key component design”.

  1. Please check the section number. Line 325 should be Section 2.2.2.

A: In line 325, I have replaced “2.2.1. Simulation test program” with “2.2.2. Simulation test program”.

  1. Please check Equation (6), the author should be more careful.

A: The errors have been corrected.

  1. Table 5 contains Chinese.

A: I have changed the Chinese in Table 5 into English.

  1. Figure 7 is confusing. The upper and lower figures seem to be the same. Where are figures (c) and (d)? Figure 7 looks messy and should be rearranged.

A: I have made a replacement for Figures 7(c) and 7 (d).